# Faster Than Flash: Exploiting Attention Sparsity for Efficient Long-Context Decoding

**Zhigeng Liu** [1 2]  **Zhiyuan Ning** [1 2]  **Ruixiao Li** [1 2]  **Xiaoran Liu** [1 2]  **Yuerong Song** [1 2]  **Min Zhang** [3 †]  **Ziwei He** [2 †]
**Xipeng Qiu** [1 2 †]

The development of long-context Large Language Models (LLMs) is constrained by the memory bandwidth bottleneck and quadratic complexity of the attention mechanism during decoding. To overcome the inherent trade-offs between the memory overhead of metadata-based metrics and the computational inefficiency of adaptive selection strategies, we present **Faster Flash Decoding (FFD)**, a novel hardware-algorithm co-design framework designed to break the memory wall in long-context decoding. FFD integrates the selector and computer into a fully fused kernel, replacing external metadata indices with content-aware scanning via low-bit quantization. Furthermore, we introduce the top-$\delta$ strategy, which dynamically filters blocks to achieve distribution-adaptive sparsity without global synchronization. Offering a training-free and plug-and-play solution, FFD also enables the reuse of scanning results for computation, achieving up to $11.6\times$ kernel-level speedup and scaling to 256K context length, with $2.37\times$ end-to-end throughput improvement. Empirical validation on RULER and LongBench confirms that FFD maintains model accuracy while delivering high-ratio sparsity, with code available at https://github.com/qluoluo/faster-flash-decoding.

## 1. Introduction

The capability to process long contexts has become a defining characteristic of modern Large Language Models (LLMs) (Achiam et al., 2023; Team et al., 2024; Dubey et al., 2024; Liu et al., 2025b). However, long-context capability comes at a prohibitive cost during the decoding phase. As the sequence length grows, the standard attention mechanism exhibits quadratic computational complexity and linear memory growth for Key-Value (KV) caches (Vaswani et al.,

[1]Fudan University, Shanghai, China [2]Shanghai Innovation Institute, Shanghai, China [3]Harbin Institute of Technology, Harbin, China. Correspondence to: Zhigeng Liu <253108120105@sii.edu.cn>, Min Zhang <zhangmin2021@hit.edu.cn>, Ziwei He <ziwei.he@sii.edu.cn>, Xipeng Qiu <xpqiu@fudan.edu.cn>.

*Proceedings of the 43$^{rd}$ International Conference on Machine Learning*, Seoul, South Korea. PMLR 306, 2026. Copyright 2026 by the author(s).

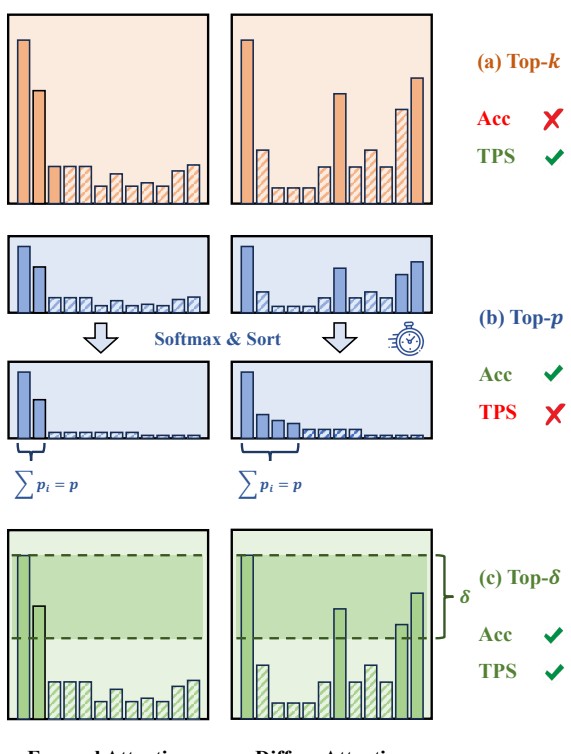

*Figure 1.* Comparison of selection strategies. (a) Top-$k$ (fixed budget) is fast but fails to adapt to distribution shape. (b) Top-$p$ offers dynamic cardinality but introduces non-trivial overhead due to global synchronization and sorting, which breaks the streaming pipeline. (c) Top-$\delta$ (ours) uses a relative threshold $\delta$ from the local max, achieving the adaptivity of top-$p$ with the hardware efficiency of top-$k$.

2017), creating a severe memory wall bottleneck in the decoding stage (Dao, 2023; Kwon et al., 2023). Unlike the prefill phase where computation dominates and queries are processed in parallel, the autoregressive decode phase is memory-bandwidth-bound: each token generation requires reloading the entire KV cache from HBM, making IO the primary bottleneck rather than FLOPs. Consequently, sparse attention mechanisms, particularly training-free, plug-and-

play approaches, have emerged as a critical research direction to reduce computational overhead without retraining the model (Tang et al., 2024; Ribar et al., 2023; Lin et al., 2025; Yuan et al., 2025; Liu et al., 2025a).

Despite the diversity of sparse attention techniques, most approaches adhere to a decoupled Selector-Computer paradigm. In this standard workflow, the selector first filters indices based on metrics, and the computer subsequently retrieves data for calculation. This separation prevents the Computer from reusing the Selector's intermediate computations, leading to efficiency bottlenecks. Furthermore, the two core components within this paradigm—the importance metric and the selection strategy—face their own dilemmas. On one hand, to determine which tokens to retain, existing methods typically rely on extra metadata, such as partial dimensions, cluster centroids, or geometric bounds (Ribar et al., 2023; Xiao et al., 2024; Tang et al., 2024). These approaches result in a metric dilemma: they either incur additional memory overhead to store these metadata structures or suffer from information distortion. On the other hand, selecting the optimal subset of tokens remains challenging. As shown in Fig 1, *top-k selection* is computationally efficient but fails to adapt to the varying entropy of attention distributions(Zhang et al., 2023; Ge et al., 2023). Comparatively, *top-p selection* offers theoretical superiority by adapting to the distribution, but needs a global softmax operation to compute cumulative probabilities, thus limiting its efficiency (Lin et al., 2025).

In this paper, we propose **FFD**, **Faster Flash Decoding**, a novel sparse attention framework that addresses these challenges through a hardware-algorithm co-design. By integrating the selector and computer, FFD enables the computation stage to directly reuse the scanning results from the selection stage. We shift from metadata-based indexing to content-aware scanning. Instead of storing extra metadata, we split the K cache into low-bit quantization and half-precision residual, taking the former as a compressed representation of the KV cache. Then, we introduce a **top-$\delta$** strategy rooted in numerical precision analysis. Rather than enforcing a fixed budget ($k$) or computing global probabilities ($p$), top-$\delta$ dynamically filters blocks based on their estimated contribution to the attention score sum and can be approximately parallelized thanks to the properties of attention in LLMs (Xiao et al., 2023). We implement FFD as a fully fused kernel that eliminates reduction bubbles, achieving $11.6\times$ kernel-level speedup. Our contributions are summarized as follows:

- **Content-Aware Scanning via Low-Bit Quantization**. We resolve the "metric dilemma" by replacing metadata-based indexing with content-aware scanning. By partitioning the K-cache into low-bit quantized representations and 8-bit residuals, we eliminate the mem-

ory overhead of extra metadata while maintaining high information fidelity.

- **The Top-$\delta$ Selection Strategy**. We introduce top-$\delta$, a novel selection mechanism, unlike top-$k$ (fixed budget) or top-$p$ (high overhead), top-$\delta$ dynamically filters attention blocks based on their contribution to the attention sum, offering distribution-adaptive sparsity with the efficiency of parallel execution.

- **Faster Flash Decoding (FFD)**. We present FFD, a hardware-algorithm co-design framework tailored for the bandwidth-critical decode phase. We implement a fused selector-computer kernel where the calculation phase reuses the selector's quantized scan, optimizing memory access patterns. Our implementation achieves up to $11.6\times$ kernel-level speedup and scales to 256K context, with up to $2.37\times$ end-to-end throughput increase while remaining a plug-and-play, training-free solution.

- **Extensive Empirical validation**. We demonstrate that FFD maintains the performance of LLMs across various short and long-context benchmarks, including RULER (Hsieh et al., 2024) and LongBench (Bai et al., 2024), proving that high-ratio sparsity can be achieved without retraining or sacrificing accuracy.

**Conflict of Interest Disclosure** We declare no financial conflicts of interest related to this work.

## 2. Related Work

**Hardware-Aware Exact Attention.** The standard attention mechanism imposes a quadratic cost during prefilling and a linear cost per step during decoding, creating a significant bottleneck for long-context applications. FlashAttention (Dao, 2023; Shah et al., 2024) revolutionized this field by introducing IO-aware optimizations, employing tiling and kernel fusion to minimize High-Bandwidth Memory (HBM) access. FlashDecoding and FlashDecoding++ (Hong et al., 2023) further adapted these principles to the generative phase by parallelizing attention computation across the sequence dimension. While FFD inherits the spirit of kernel fusion and hardware-algorithm co-design from these works, existing Flash-based methods fundamentally remain *exact* attention mechanisms. As context length increases, they must still load and scan the entire KV cache, resulting in unavoidable linear growth in latency. FFD addresses this by integrating sparsity directly into the fused kernel, breaking the linear scanning barrier while preserving hardware efficiency.

**Dynamic Sparse Attention** To circumvent the costs of exact attention, various sparse attention mechanisms have

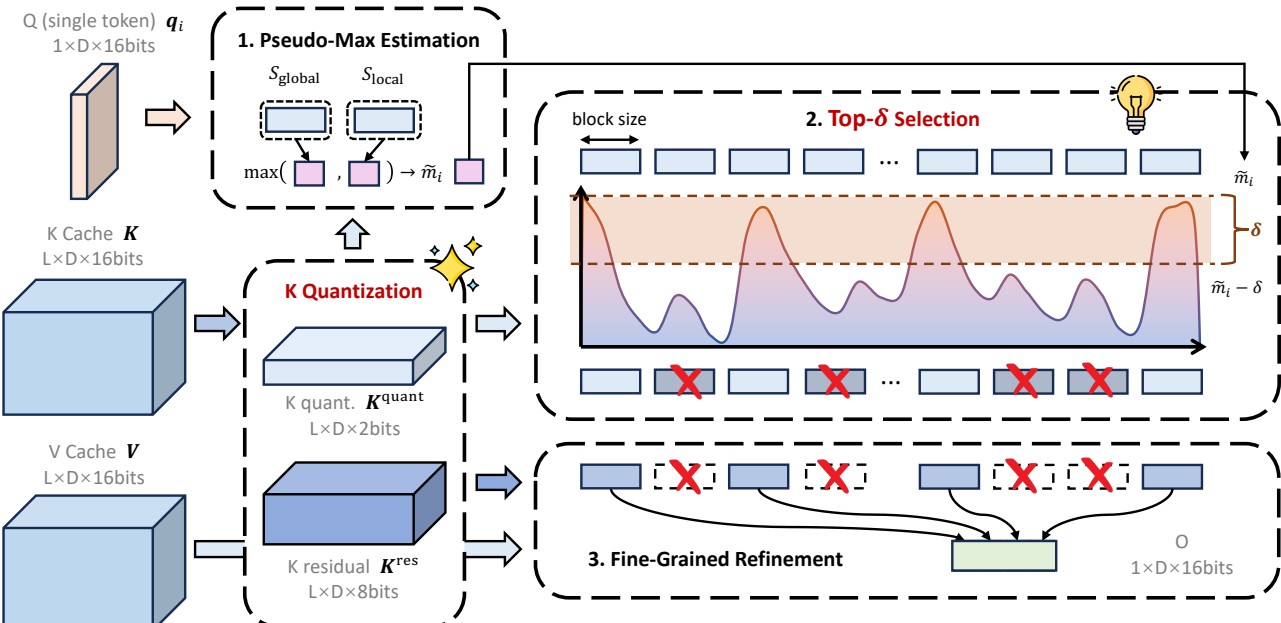

*Figure 2.* Overview of our FFD implementation. Inputs include the 16-bit query as well as the value cache and the key cache split into a 2-bit quantized thumbnail and an 8-bit residual. The attention output is acquired after K quantization, pseudo-max estimation, top-$\delta$ selection, as well as filtering based on the top-$\delta$ strategy, and fine-grained refinement.

been proposed. These methods generally face specific challenges in importance estimation (the Metric Dilemma) and subset filtering (the Selection Dilemma). Regarding the Metric Dilemma, methods like H2O (Zhang et al., 2023), Scissorhands (Liu et al., 2023), and SnapKV (Li et al., 2024) rely on historical accumulation to identify "heavy hitters" for token eviction. However, purely history-based metrics risk discarding tokens that become relevant only in future contexts. To address this, dynamic retrieval methods such as SparQ (Ribar et al., 2023) and Quest (Tang et al., 2024) employ auxiliary metadata (e.g., mean vectors or Min-Max bounds) to estimate block importance. Similarly, InfLLM (Xiao et al., 2024) utilizes representative blocks for neighborhood search. While these metadata-based approaches reduce scanning costs, they suffer from information distortion and require additional memory management. Regarding the Selection Dilemma, most approaches (e.g., Quest, SparQ) enforce a static *top-k* budget, which lacks flexibility across varying attention entropy distributions. Twilight (Lin et al., 2025) attempts to adapt to various attention distributions but introduces a global softmax operation, thus limiting its efficiency. Concurrent to our work, SALE (Ji et al., 2025) independently explored low-bit estimation with anchor-based selection for sparse attention. However, SALE targets the prefill stage and employs 4-bit quantization, doubling the scanning bandwidth compared to our 2-bit design tailored for bandwidth-critical decoding. In contrast, FFD performs content-aware scanning via a fused

Selector-Computer kernel, bypassing the need for auxiliary structures and enabling a dynamic *top-$\delta$* selection strategy that adapts to entropy changes on the fly.

**KV Cache Quantization** KV cache quantization is primarily employed to alleviate memory capacity constraints. Methods such as KIVI (Liu et al., 2024), Atom (Zhao et al., 2024), and KVQuant (Hooper et al., 2024) compress keys and values into low-precision formats to fit longer contexts into GPU memory. Typically, these frameworks dequantize data back to higher precision (FP16/BF16) before performing the attention computation to preserve accuracy. FFD fundamentally repurposes quantization. Instead of treating low-bit representations solely as a storage compression technique, we utilize the quantized INT2/4 data as a high-speed proxy for the importance metric itself. This allows our selector to scan the full context with high fidelity and minimal latency, avoiding the decompression overhead typical of standard quantization frameworks while maintaining superior precision compared to metadata-based heuristics.

## 3. Method

In this section, we present **FFD** (Faster Flash Decoding), a hardware-aware sparse attention framework designed to break the memory wall in long-context decoding. FFD is built upon a hardware-algorithm co-design that systematically resolves the trade-offs in sparse attention through

two core innovations: Content-Aware Scanning via Low-Bit Quantization and top-$\delta$ Selection Strategy. FFD utilizes low-bit quantization thumbnails of the K cache to perform high-fidelity importance estimation without additional memory overhead. To adapt to varying attention distributions, the top-$\delta$ criterion is introduced to provide a theoretically grounded, parallelizable alternative to the rigid top-$k$ or computationally expensive top-$p$ selection. We implement a fused operator for the selector and computer to optimize the execution; specifically, the computer reuses the selector's estimation results to continue the computation, thereby accelerating the process while efficiently supporting the adaptive selection. The overall procedure is summarized in Figure 2 and Alg. 1.

The remainder of this section is organized as follows. First, we detail the efficient quantization-based scanning mechanism that enables high-fidelity importance estimation. Next, we introduce the formulation and parallel implementation of the top-$\delta$ selection. Finally we describe the system-level optimization and fused kernel design that underpin the high-speed execution of FFD.

### 3.1. Content-Aware Scanning via Low-Bit Quantization

Inspired by advances in KV cache quantization (Liu et al., 2024; Hooper et al., 2024; Yang et al., 2024), we decouple retrieval fidelity from storage overhead by decomposing the K cache as $\boldsymbol{k} \to \{\boldsymbol{k}^{\text{quant}}, \boldsymbol{k}^{\text{res}}\}$. Here, $\boldsymbol{k}^{\text{quant}}$ represents 2-bit quantized thumbnails utilized for high-throughput similarity scanning, while $\boldsymbol{k}^{\text{res}}$ denotes 8-bit residuals.

To maximize information entropy within the limited bit-width, we employ a symmetric zero-free (mid-rise) quantization scheme, which ensures that directional information is preserved even for small-magnitude features (details in Appendix A).

This architecture ensures that the scanning phase operates at minimal arithmetic intensity, while the identified candidates are reconstructed to near-FP16 precision for exact computation. A formal concentration bound on the scanning error is provided in Appendix B.

### 3.2. Top-$\delta$ Selection Criterion

Let $s_{ij} = \dfrac{\boldsymbol{q}_i^\top \boldsymbol{k}_j}{\sqrt{d}}$ be the pre-softmax attention score between query vector $\boldsymbol{q}_i$ and key vector $\boldsymbol{k}_j$. Ideally, we aim to retain only those tokens whose attention weight contributes significantly to the distribution. We define a retention condition based on the relative magnitude $\delta$ compared to the maximum attention score $m_i = \max_j s_{ij}$, and thus only calculate the attention over the KV cache where

$$s_{ij} \geq m_i - \delta. \tag{1}$$

---

**Algorithm 1** FFD Decoding with Hybrid Quantization

1: **Input:** Query $\boldsymbol{q}_i$, 2-bit Keys $\boldsymbol{K}^{\text{quant}}$, 8-bit Residuals $\boldsymbol{K}^{\text{res}}$, Values $\boldsymbol{V}$, threshold $\delta$, sink indices $S_{\text{global}}$, local indices $S_{\text{local}}$.
2: **Output:** Attention output $o_t$
3: # Pseudo-max estimation
4: $s_{\text{global}} \leftarrow \max_{j \in S_{\text{global}}} \left( \boldsymbol{q}_i^\top \cdot \boldsymbol{k}_j^{\text{quant}} \right)$
5: $s_{\text{local}} \leftarrow \max_{j \in S_{\text{local}}} \left( \boldsymbol{q}_i^\top \cdot \boldsymbol{k}_j^{\text{quant}} \right)$
6: $\tilde{m}_i \leftarrow \max \left( s_{\text{global}}, s_{\text{local}} \right)$
7:
8: # Top-$\delta$ selection
9: $\mathcal{I}_{\text{selected}} \leftarrow \emptyset$
10: **for** each block index $B$ **do**
11: $\quad s^{\text{approx}} \leftarrow \boldsymbol{q}_i \cdot \boldsymbol{K}^{\text{quant}}[B]$
12: $\quad$ **if** $\max(s^{\text{approx}}) \geq \tilde{m}_i - \delta$ **then**
13: $\quad\quad \mathcal{I}_{\text{selected}} \leftarrow \mathcal{I}_{\text{selected}} \cup B$
14: $\quad$ **end if**
15: **end for**
16:
17: # Fine-grained refinement
18: logits $\leftarrow []$
19: **for** each block index $B$ **do**
20: $\quad \boldsymbol{K}^{\text{full}} \leftarrow \text{Dequant}(\boldsymbol{K}^{\text{quant}}[B], \boldsymbol{K}^{\text{res}}[B])$
21: $\quad s_{\text{final}} \leftarrow \dfrac{\boldsymbol{q}_i \cdot \boldsymbol{K}^{\text{full}}}{\sqrt{d}}$
22: $\quad$ logits.append($s_{\text{final}}$)
23: **end for**
24: $o_t \leftarrow \text{Softmax}(\text{logits}) \cdot V[\mathcal{I}_{\text{selected}}]$

---

This additive threshold in the log-space, $\delta$, translates to a rigorous multiplicative bound in the probability space. By exponentiating Eq. 1, we observe that a retained token $j$ must satisfy

$$\frac{\exp(s_{ij})}{\exp(m_i)} \geq e^{-\delta}. \tag{2}$$

This provides a clear physical interpretation that $\delta$ controls the maximum allowable attention drop-off. For instance, setting $\delta = 5$ guarantees that we discard only tokens whose contribution is less than $e^{-5} \approx 0.67\%$ of the peak attention mass. This allows our FFD to adaptively vary the retrieval budget based on the sharpness of the attention distribution, rather than a fixed top-$k$ and top-$p$ as shown in Fig. 1.

Implementing Eq. 1 strictly requires computing the global max $m_i$, which incurs the same synchronization overhead as softmax. Fortunately, thanks to the phenomenon that attention distributions are typically dominated by either local context or initial sink tokens (Xiao et al., 2023), we introduce the pseudo-max approximation $\tilde{m}_i$ by estimating the global maximum using only these accessible subsets,

sink tokens $S_{\text{global}}$ and local context $S_{\text{local}}$, as follows.

$$\tilde{m}_i = \max \left( \max_{t \in S_{\text{global}}} s_{it}, \max_{t \in S_{\text{local}}} s_{it} \right). \quad (3)$$

This approximation allows the threshold $\tilde{m}_i - \delta$ to be computed using only locally available data, effectively bypassing the global reduction bottleneck and enabling independent parallel processing of KV blocks.

### 3.3. Kernel Optimization

We implement FFD as a cooperative pipeline comprising three specialized Triton kernels, illustrated in Fig. 2. The execution flow proceeds as follows:

1. **Pseudo-max estimation**. A lightweight kernel computes the pseudo-max $\tilde{m}$ using only the sink and local tokens.

2. **Top-$\delta$ selection**. The main kernel loads 2-bit keys in streaming chunks. It computes tentative scores and compares them against $\tilde{m} - \delta$.

3. **Fine-grained refinement**. If a block passes the filter, the kernel loads the corresponding 8-bit residual keys and values.

The add-on computation in the refinement stage is shown as follows and ensures that for selected tokens, the attention score is computed with near-FP16 precision.

$$s_{\text{final}} = \frac{\boldsymbol{q}^\top \boldsymbol{k}^{\text{quant}} + \boldsymbol{q}^\top \boldsymbol{k}^{\text{res}}}{\sqrt{d}} \quad (4)$$

This two-term form is semantically equivalent to the `Dequant` step in Algorithm 1 but corresponds more closely to the actual kernel implementation, where the two dot products are accumulated without materializing a full-precision key tensor.

To further minimize latency, particularly for decoding with small batch sizes where CPU launch overhead dominates, we implement a full-chain CUDA graph strategy. Unlike standard implementations that only capture the attention kernel, we capture the entire decoding step, including MLP, RMSNorm, and residual connections.

However, standard CUDA Graphs are static and incompatible with the dynamic sequence growth of KV caches. We address this with two innovations:

- **Block-wise JIT capture**. We introduce a dynamic graph capture mechanism that triggers re-capture only when the number of full KV blocks changes. This amortizes compilation cost over the block size.

- **Graph-friendly cache**. We replace standard Python slicing (which triggers CPU-GPU synchronization)

with tensor-based indexing kernels, ensuring the entire cache update process remains device-side and graph-capturable.

This hybrid execution model runs 99% of steps within a CUDA Graph, falling back to eager mode only at block boundaries for memory allocation.

## 4. Experiments

### 4.1. Setup

**Device** We conduct evaluations on two distinct GPU architectures, NVIDIA H100, representing datacenter-grade hardware with high memory bandwidth, and NVIDIA GeForce RTX 4090, representing consumer-grade hardware. All experiments are implemented using PyTorch 2.8 and Triton 3.4 under CUDA 12.8.

**Baselines** We compare FFD with full attention accelerated by FlashAttention-2 (Dao, 2023), Quest (Tang et al., 2024), KIVI (Liu et al., 2024), and Twilight (Lin et al., 2025). Twilight does not provide fully open-source official code at the time of writing; its results are based on our own faithful reproduction following the algorithm described in the original paper. To ensure a fair system-level comparison, we align the token budget as 16k and memory access patterns with hardware characteristics. For Quest, we utilize the official page size of $P = 16$. Fortunately, under this configuration, the memory footprint of the indices among these methods is equivalent to that of our 2-bit data for full scanning, which ensures a fair comparison.

**Hyperparameter** For FFD, we evaluate sparsity thresholds $\delta \in \{5, 7\}$. These values are selected based on their theoretical implications for the probability mass:

- $\delta = 5$ implies pruning tokens whose attention weight $\exp(s_{ij})$ is less than $e^{-\delta} \approx 1/148$ of the maximum score. This represents an aggressive filtering strategy targeting the top $\sim 1\%$ relevant tokens.

- $\delta = 7$ corresponds to a threshold of $e^{-7} \approx 1/1096$, serving as a conservative setting that retains tokens with even $\sim 0.1\%$ relative importance to prevent tail-risk information loss.

### Evaluation

We structure our evaluation into two parts: Efficiency and Effectiveness.

- **Efficiency:** We evaluate system performance at two levels. We use *Trace-Driven Kernel Microbenchmarks* (Section 4.2) to measure operator latency, and *End-to-End Generation* (Section 4.3) to test the throughput of the full model.

- **Effectiveness:** We validate that our method preserves model quality. We employ standard long-context benchmarks, specifically LongBench and RULER (Section 4.4), to assess performance on downstream tasks.

## 4.2. Kernel Efficiency

We benchmark the latency of our fused Triton kernel on a consumer GPU with sequence lengths ranging from 4K to 256K. To minimize launch overheads in the low-latency regime (¡1ms), we utilize CUDA Graphs for kernel submission. We compare against the standard PyTorch `scaled_dot_product_attention` (which utilizes FlashAttention-2).

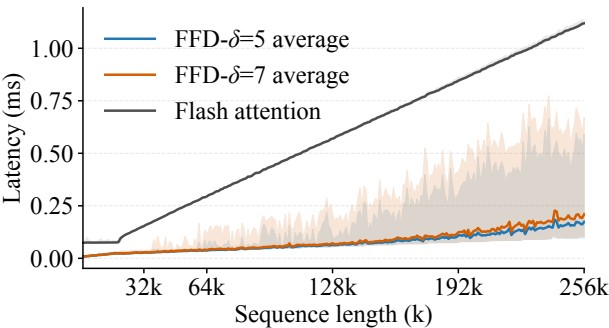

*Figure 3.* Kernel Latency vs. Sequence Length (RTX 4090). Layer-averaged execution time for FlashAttention-2 and FFD variants (batch=1). FFD-$\delta = 5$ and $\delta = 7$ achieve average speedups of $7.33\times$ and $6.18\times$ respectively, with a peak reduction of $11.63\times$. Error bands denote the min–max range across layers.

As shown in Figure 3, our FFD kernel demonstrates linear scaling with sequence length but with a significantly lower slope; error bands indicate the layer-wise min–max range for FFD ($\delta = 5$, $\delta = 7$) and FlashAttention-2. At 256K context length, FlashAttention-2 averages 1.12 ms, while FFD ($\delta = 5$) and FFD ($\delta = 7$) average 0.17 ms and 0.21 ms, corresponding to $6.58\times$ and $5.33\times$ speedups. This dramatic acceleration is achieved by combining: 1. *2-bit IO:* Reducing memory access from 16 bits to 2 bits for the scanning phase. 2. *High Sparsity:* The $\delta = 5$ threshold effectively filters a lot of blocks on real Llama-3.1 data. 3. *CUDA Graphs:* Eliminating CPU launch overhead, which is critical when the kernel runtime is sub-100$\mu$s.

## 4.3. End-to-End Throughput

Beyond kernel-level microbenchmarks, we evaluate the impact of FFD on end-to-end token generation throughput. Figure 4 compares the maximum throughput (tokens/sec) achievable by FFD, Quest, FlashAttention-2, and KIVI across varying context lengths on both NVIDIA RTX 4090 and H100 GPUs.

On the consumer-grade RTX 4090, FFD consistently achieves the highest throughput, reaching 61.0 tokens/s at 4K and maintaining 51.8 tokens/s at 16K context. This represents a speedup of up to $2.37\times$ over FlashAttention-2 (at 16K). Quest also performs well but lags behind FFD (e.g., 47.8 vs 51.8 at 16K).

We further scale our evaluation to the datacenter-grade H100. FFD demonstrates strong scalability, reaching 110.6 tokens/s at 4K context and maintaining 87.0 tokens/s at 16K. This corresponds to a $1.96\times$ speedup over FlashAttention-2 (44.5 tokens/s) at 16K context. Consistent with 4090 results, KIVI continues to underperform dense attention on H100 in this low-batch regime (e.g., 22.4 vs 44.5 at 16K), highlighting the significant cost of dequantization.

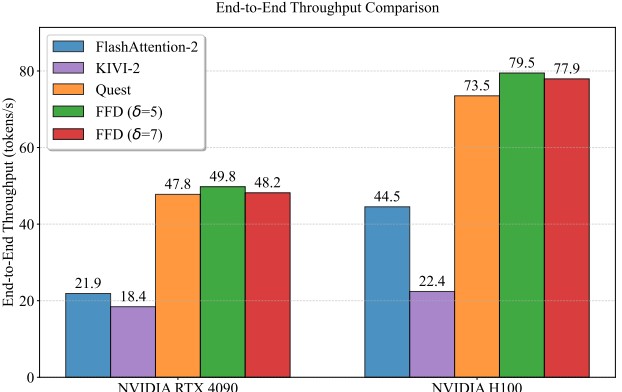

*Figure 4.* End-to-End Generation Throughput Comparison on RTX 4090 and H100 (16K Context). FFD outperforms FlashAttention-2, Quest, and KIVI on both platforms. On H100, FFD achieves up to $1.96\times$ speedup over the dense baseline and surpasses Quest (e.g., 87.0 vs 73.5 tokens/s at 16K). KIVI suffers from dequantization overheads in this single-batch setting.

## 4.4. Downstream Performance

To validate the effectiveness of FFD in real-world long-context scenarios, we evaluated our method on the RULER benchmark (Hsieh et al., 2024) with a context length of 32K. Table 1 presents the results comparing FFD against the dense Llama-3.1-8B-Instruct baseline.

Our method maintains high performance across all categories. Notably, on the challenging "Needle In A Haystack" (NIAH) Single and Multi-Key retrieval tasks, FFD achieves near-perfect scores (e.g., 100.0 on SK-1/SK-2, 99.0 on SK-3), matching the dense baseline. Even on complex reasoning tasks like Variable Tracking (VT), the performance degradation is minimal (98.4 vs 99.6), demonstrating that our 2-bit high-fidelity filter successfully preserves the critical information required for multi-hop reasoning.

We further evaluate FFD on LongBench (Bai et al., 2024), a

*Table 1.* RULER Benchmark Results (32k Context). Comparison of Llama-3.1-8B-Instruct Base, StreamingLLM, KIVI (2-bit, group=32), Quest (ratio=0.5), Twilight, and FFD. We report the aggregated score (0–100) for each task category. **Bold** indicates the best performance among sparse methods (excluding Base). Abbreviations: SK (SingleKey), MK (MultiKey), MQ (MultiQuery), MV (MultiValue), VT (Variable Tracking), CWE (Common Words Extraction), FWE (Frequent Words Extraction). Avg is the simple average over all 13 RULER subtasks.

| Method | SK-1 | SK-2 | SK-3 | MK-1 | MK-2 | MK-3 | MQ | MV | VT | SQuAD | Hotpot | CWE | FWE | Avg |
|---|---|---|---|---|---|---|---|---|---|---|---|---|---|---|
| Base | 100.0 | 100.0 | 100.0 | 98.0 | 100.0 | 99.0 | 98.5 | 98.5 | 99.6 | 67.0 | 56.0 | 67.8 | 93.0 | 90.6 |
| StreamingLLM | 3.0 | 0.0 | 0.0 | 2.0 | 2.0 | 1.0 | 0.0 | 0.0 | 0.0 | 11.0 | 9.0 | 0.2 | 51.0 | 6.1 |
| KIVI | **100.0** | 99.0 | 98.0 | 93.0 | 93.0 | 48.0 | 86.8 | 86.0 | 99.2 | 63.0 | **54.0** | 56.4 | **93.3** | 82.3 |
| Quest (0.5) | **100.0** | **100.0** | **100.0** | **96.0** | 64.0 | 6.0 | 98.3 | 96.3 | **99.4** | 65.0 | 49.0 | 15.8 | 71.0 | 73.9 |
| Twilight | **100.0** | 88.0 | 96.0 | 88.0 | 76.0 | 37.0 | 56.5 | 60.3 | 98.2 | 60.0 | 49.0 | 12.7 | 86.3 | 69.8 |
| FFD ($\delta = 5$) | **100.0** | 99.0 | 99.0 | **96.0** | 97.0 | 78.0 | 98.3 | 97.8 | 96.8 | 65.0 | 52.0 | 61.0 | 93.0 | 87.1 |
| FFD ($\delta = 7$) | **100.0** | **100.0** | 99.0 | **96.0** | **98.0** | **95.0** | **99.5** | **98.5** | 98.4 | **66.0** | 52.0 | 67.0 | 92.7 | **89.4** |

*Table 2.* LongBench Results. Comparison of Base, StreamingLLM, KIVI (2-bit), Quest, Twilight, and FFD. We report the average score (%) for each category. **Bold** indicates the best performance among sparse methods (excluding Base). Twilight results are based on our own reproduction.

| METHOD | S-DOC | M-DOC | SUMM | FEW-SHOT | SYN | CODE | AVG |
|---|---|---|---|---|---|---|---|
| BASE | 24.02 | 15.24 | 16.57 | 44.10 | 32.70 | 24.68 | 26.22 |
| STREAMINGLLM | 3.44 | 3.12 | 5.92 | 22.49 | 0.02 | 22.55 | 9.59 |
| KIVI | 23.46 | 14.44 | 15.46 | 44.17 | 30.48 | 24.94 | 25.49 |
| QUEST | 23.13 | 14.33 | 15.29 | 43.69 | 30.28 | **29.36** | 26.01 |
| TWILIGHT | 23.25 | 15.53 | **16.97** | 43.73 | **32.37** | 25.23 | 26.18 |
| FFD ($\delta = 5$) | 23.36 | 15.35 | 16.40 | 44.09 | 30.04 | 25.94 | 25.86 |
| FFD ($\delta = 7$) | **24.00** | **15.78** | 16.11 | **44.19** | 31.83 | 26.21 | **26.35** |

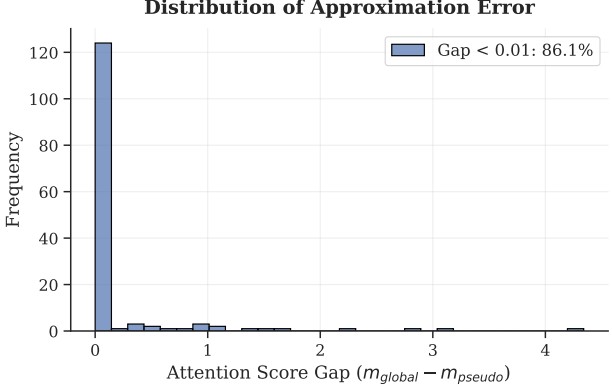

**Distribution of Approximation Error**

*Figure 5.* Distribution of the approximation error ($m_i - \tilde{m}_i$) by layer. The gap is effectively zero for most of conditions, confirming that the maximum attention score is dominated by sinks and local tokens. This verifies that our Pseudo-Max approximation serves as a robust safe relaxation.

comprehensive benchmark for long-context understanding containing 21 datasets across 6 categories. Table 2 reports the category-level results across all compared methods.

FFD demonstrates strong performance across all six categories. With an overall average of 26.35, FFD ($\delta = 7$) achieves the highest aggregate score among all methods,

reinforcing our hypothesis that 2-bit content-aware scanning preserves the semantic fidelity required for diverse long-context understanding tasks.

**Comparison with Top-$p$ Selection.** To provide a direct comparison with the Top-$p$ selection paradigm, we additionally benchmark Twilight (Lin et al., 2025) under its original operating point (budget=8192, $p = 0.85$). As shown in Tables 1 and 2, FFD consistently outperforms Twilight by substantial margins on both benchmarks, while maintaining a clear advantage over Quest. The multi-peak retrieval tasks (MK-2, MK-3) particularly highlight the weakness of fixed-budget selectors: Quest drops sharply on MK-3 (6.0), whereas FFD with top-$\delta$ adapts to the distributed attention pattern (95.0). We omit throughput comparison with Twilight because its official implementation is not fully open-source, making a fair system-level speed comparison infeasible.

### 4.5. Ablation Study

To disentangle the contributions of our three co-design axes—scan precision (FP16 vs. 2-bit), selection rule (top-$k$ vs. top-$\delta$), and execution path (split vs. fused+fullgraph)—we conduct a controlled ablation study under matched conditions. All variants are evaluated with a 16K prefill and 128 decode steps on an RTX 4090 (batch size 1). The full ablation table and detailed analysis are presented in Ap-

*Table 3.* Per-head selection latency at matched keep ratio (∼27.3%).

| SELECTION RULE | KEEP RATIO | LATENCY (MS/HEAD) |
| --- | --- | --- |
| TOP-$\delta$ | 0.273 | 0.0044 |
| TOP-$k$ | 0.275 | 0.0090 |
| TOP-$p$ | 0.273 | 0.1115 |
| KIVI2 (DENSE) | 1.000 | 0.0186 |

pendix G. In summary, the fused 2-bit top-$\delta$ configuration (FFD) achieves 53.96 tok/s with RULER AVG of 90.40 and LongBench AVG of 25.90—a synergistic 2.2× throughput gain over the split-execution counterpart—demonstrating that the algorithmic and system designs are inseparably linked.

**Selection-Rule Overhead.** To isolate the algorithmic cost of each selection rule, we calibrate top-$k$ and top-$p$ to match the average keep ratio of top-$\delta$ (∼27.3%). Table 3 reports the per-head latency breakdown under matched sparsity. At nearly identical keep ratios, top-$\delta$ is 2× faster than top-$k$ and 25× faster than top-$p$, confirming that the top-$\delta$ formulation eliminates the global synchronization bottleneck inherent to cumulative probability methods.

**Generalization to Qwen2.5.** We further validate FFD on Qwen2.5-7B-Instruct to assess generalization beyond the Llama architecture. Under the same $\delta = 7$ configuration, FFD achieves 85.90 RULER AVG and 30.11 LongBench AVG, outperforming KIVI (72.01 / 28.34), Quest (71.93 / 29.49), and StreamingLLM (16.24 / 21.45). These results confirm that FFD's content-aware scanning and top-$\delta$ selection generalize effectively across model families without architecture-specific tuning.

## 5. Discussion

### 5.1. Retrieval Fidelity Analysis

We evaluate the "quality" of the retrieved tokens by measuring two metrics:

1. *Recall:* The cosine similarity between the sparse attention output and the dense (exact) output.

2. *LSE Error:* The Max Absolute Error of the Log-Sum-Exp (LSE) value, which proxies the stability of the Softmax distribution.

FFD achieves significantly higher recall than Quest across all sparsity ratios, confirming the "Thumbnail vs Bounding Box" hypothesis: 2-bit quantization preserves the geometric directionality of keys, enabling more precise filtering than min/max bounds. FFD also minimizes the error in the normalization term (LSE), which is critical for preventing "collapse" in the attention distribution.

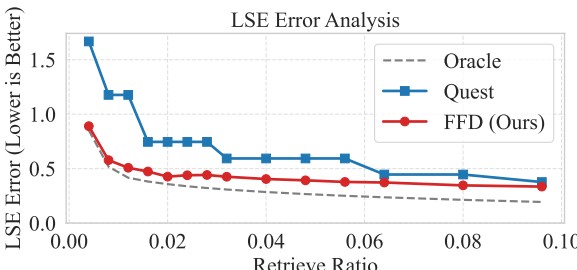

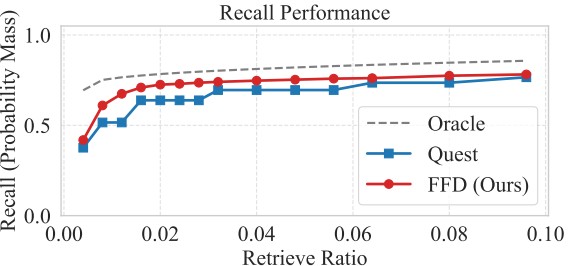

*Figure 6.* Quality vs. IO Budget on Llama. FFD maintains lower LSE error while achieving higher recall than Quest under the same retrieval budget, showing that high-fidelity scanning preserves distributional accuracy and hit rate.

Figure 7 shows that FFD consistently maintains higher Spearman correlation than Quest across all layers, indicating that quantization-based scanning better preserves the relative ordering of attention scores, whereas the bounding-box approximation suffers from degradation.

### 5.2. Pseudo-Max Robustness

FFD avoids a global reduction by estimating the threshold using a Pseudo-Max computed from sink and local tokens. Figure 5 reports the empirical gap between the global maximum and the Pseudo-Max in a simulation that mirrors the sink/local bias.

We analyze the implications of the approximation gap where $\tilde{m}_i < m_i$. As shown in Figure 5, such gaps are rare and numerically negligible. Crucially, even when they occur, this underestimation represents a fail-safe mechanism. Recall our filtering condition: $s_{ij} \geq \tilde{m}_i - \delta$. Since $\tilde{m}_i \leq m_i$ by definition, it follows that our threshold is lower than the ideal threshold: $\tilde{m}_i - \delta \leq m_i - \delta$. Consequently, any token that would be selected by the global max is *guaranteed* to be selected by the Pseudo-Max. In these rare "failure" cases, FFD automatically falls back to a more conservative retrieval policy, preserving slightly more tokens. This structural asymmetry ensures that approximation errors result only in a minor increase in I/O budget (efficiency penalty), rather than the loss of critical information (accu-

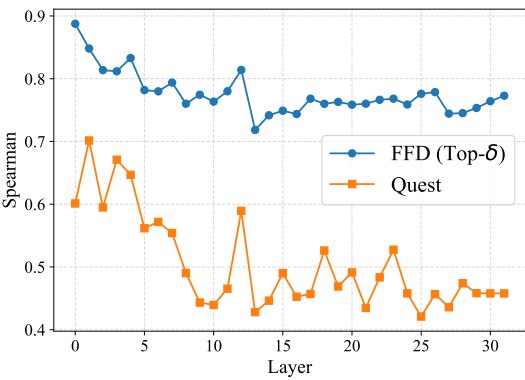

*Figure 7.* Layer-wise Correlation Trends. FFD achieves consistently higher Spearman correlations across all layers compared to Quest, indicating better preservation of the attention distribution's structural properties.

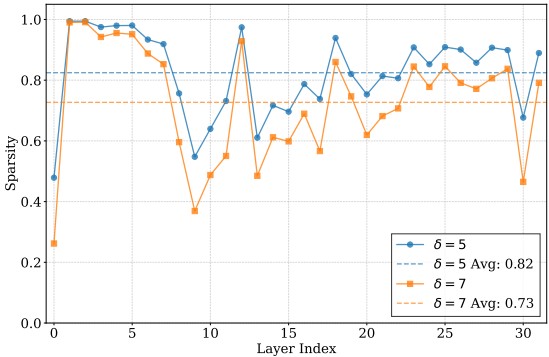

*Figure 8.* Layer-wise sparsity distribution on LongBench (Llama-3.1-8B). FFD achieves significant computational savings, maintaining average sparsity levels of 82% and 73% under $\delta = 5$ and $\delta = 7$ configurations respectively.

racy penalty).

### 5.3. Sparsity Analysis

FFD exhibits layer-wise adaptive sparsity governed by $\delta$, with higher sparsity at smaller $\delta$. As shown in Figure 8, FFD maintains average sparsity of 82% and 73% under $\delta = 5$ and $\delta = 7$ respectively—operating with substantially lower compute budget than dense attention—while Tables 1 and 2 confirm no accuracy sacrifice. By mitigating redundant I/O overhead, FFD achieves superior inference speed compared to other approaches.

### 5.4. Limitations

We identify several limitations that point to future work. First, while our head-wise variation analysis (Appendix D) indicates that FFD adapts naturally to diverse per-head sparsity patterns, the characterization is based on controlled

single-sample statistics and has not been systematically validated across diverse task distributions. Second, the failure-case profiling in Appendix E uses proxy metrics (top-1/top-2 gap and score standard deviation) rather than direct end-to-end task-failure attribution; a more comprehensive causal attribution remains open. Third, FFD has been validated on Llama and Qwen architectures employing grouped-query attention (GQA); its applicability to MLA-style architectures (Liu et al., 2025a) with fundamentally different key/value factorization has not yet been assessed. Finally, while our component-wise ablation isolates the major contributors to the end-to-end speedup, a fine-grained latency breakdown within the fused execution path is deferred to future work.

## 6. Conclusion

We presented FFD, a hardware-algorithm co-design that rethinks sparse attention as geometric filtering rather than metadata indexing. By utilizing 2-bit quantization for high-fidelity scanning and attention sinks for adaptive thresholding, we break the dependency on rigid top-$k$ budgets. Our results suggest that future long-context inference should prioritize "compute-for-IO" trade-offs—spending cheap FLOPs on low-bit scanning to save expensive HBM bandwidth.

## Acknowledgments

This work was supported by the Science and Technology Commission of Shanghai Municipality (No. 25DZ3100402).

## Impact Statement

This work improves the decoding efficiency of large language models through algorithmic and kernel optimization. It does not involve human subjects, personal information, or sensitive content, and we identify no extra ethical concerns specific to our work.

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

# A. Details of Quantization Implementation

FFD employs a Symmetric Mid-rise Uniform Quantization scheme. Unlike standard integer mapping (e.g., $\{-1, 0, 1\}$) which wastes quantization bins on a zero state, our approach utilizes a strictly non-zero grid to maximize the information capacity of the limited 2-bit budget.

## A.1. Formulation

We define a 4-level quantization codebook $\mathcal{C}$ centered around zero but excluding zero itself:

$$\mathcal{C} = \{-1.5, -0.5, +0.5, +1.5\} \tag{5}$$

For a given input channel vector $\mathbf{x}$, we compute the scaling factor $s$ based on the absolute maximum value to ensure coverage:

$$s = \frac{\max(|\mathbf{x}|)}{1.5} \tag{6}$$

The quantized representation $\mathbf{x}_q$ is obtained by projecting the scaled input onto the nearest neighbor in the codebook:

$$\mathbf{x}_q = \arg\min_{c \in \mathcal{C}} \left| \frac{\mathbf{x}}{s} - c \right| \tag{7}$$

The reconstructed approximation is then $\hat{\mathbf{x}} = \mathbf{x}_q \cdot s$.

## A.2. Design Rationale

We explicitly choose a mid-rise quantizer (no zero point) over a mid-tread quantizer to align with the distributional properties of modern LLMs. Key vectors, particularly after Rotary Positional Embeddings (RoPE), are typically dense with magnitude distributed across dimensions. A standard zero-inclusive grid effectively utilizes only 3 states ($\log_2 3 \approx 1.58$ bits), wasting roughly 20% of the representational capacity of a 2-bit system. By forcing every dimension to take a non-zero value, our mid-rise scheme ensures full bit utilization and prevents "information collapse", preserving the directional fidelity required for high-dimensional dot-product scanning.

# B. Error Analysis of Quantized Scanning

In this section, we provide a theoretical bound on the approximation error introduced by the 2-bit quantization during the scanning phase. We demonstrate that as the head dimension $d$ increases, the probability of the approximation error exceeding the selection safety margin decays exponentially.

## B.1. Quantization Error Bound

Let $\mathbf{q}, \mathbf{k} \in \mathbb{R}^d$ denote the query and key vectors of dimension $d$ (head dimension). In our FFD framework, the exact pre-softmax attention score is defined as $s = \mathbf{q}^\top \mathbf{k} / \sqrt{d}$. The scanner approximates this using the 2-bit quantized key $\hat{\mathbf{k}} = \mathbf{k} + \epsilon$, where $\epsilon \in \mathbb{R}^d$ is the quantization noise vector. The approximate score is given by:

$$\hat{s} = \frac{\mathbf{q}^\top (\mathbf{k} + \epsilon)}{\sqrt{d}} = s + \frac{\mathbf{q}^\top \epsilon}{\sqrt{d}} \tag{8}$$

The approximation error is thus $e_{approx} = \frac{\mathbf{q}^\top \epsilon}{\sqrt{d}}$.

**Assumption B.1** (Sub-Gaussian Quantization Noise). While standard quantization theory models noise as uniform for high bit-depths, for 2-bit quantization, we adopt a more robust assumption: the noise components $\epsilon_i$ are independent, mean-zero, and $\sigma^2$-sub-Gaussian. This means $\mathbb{E}[\exp(t\epsilon_i)] \leq \exp(\sigma^2 t^2 / 2)$ for all $t \in \mathbb{R}$. For a symmetric 4-level quantizer with step size $\Delta$, the variance and sub-Gaussian parameter are determined by the bin width $\Delta \approx \max(|\mathbf{k}|)/1.5$.

**Theorem B.2** (Concentration of Scanning Error). *Let the query vector $\mathbf{q}$ have a bounded $\ell_2$-norm such that $\|\mathbf{q}\|_2 \leq \alpha \sqrt{d}$, where $\alpha$ is a constant determined by the model's normalization layers (e.g., RMSNorm). Under Assumption B.1, the probability that the magnitude of the scanning error $|e_{approx}|$ exceeds a deviation $\lambda$ is bounded by:*

$$P(|e_{approx}| \geq \lambda) \leq 2 \exp\left( -\frac{\lambda^2}{2\sigma^2 \alpha^2} \right) \tag{9}$$

*Proof.* The error term $e_{approx}$ is a linear combination of independent sub-Gaussian random variables:

$$e_{approx} = \sum_{i=1}^{d} \left( \frac{q_i}{\sqrt{d}} \right) \epsilon_i \tag{10}$$

The sum of independent sub-Gaussian variables $X = \sum w_i \epsilon_i$ is itself sub-Gaussian with parameter $\|\mathbf{w}\|_2^2 \sigma^2$. Here, the weights are $w_i = q_i/\sqrt{d}$. The effective sub-Gaussian parameter for $e_{approx}$ is:

$$\sigma_{eff}^2 = \sum_{i=1}^{d} \left( \frac{q_i}{\sqrt{d}} \right)^2 \sigma^2 = \frac{\sigma^2}{d} \|\mathbf{q}\|_2^2 \tag{11}$$

Using the $\|\mathbf{q}\|_2 \leq \alpha \sqrt{d}$ bound (where typically $\alpha \approx 1$ in normalized LLMs), we have $\sigma_{eff}^2 \leq \sigma^2 \alpha^2$. Applying the general Hoeffding-type bound for sub-Gaussian variables:

$$P(|e_{approx}| \geq \lambda) \leq 2 \exp \left( -\frac{\lambda^2}{2\sigma_{eff}^2} \right) \leq 2 \exp \left( -\frac{\lambda^2}{2\sigma^2 \alpha^2} \right) \tag{12}$$
$$\square$$

### B.2. Reliability of Top-$\delta$ Selection

Based on the error bound derived in Theorem B.2, we can now formally justify the FFD selection strategy. The primary goal is to minimize the False Negative Rate (FNR)—missing a token that is truly significant.

Let $\delta_{ideal}$ be the theoretically optimal threshold in log-space (Eq. 1). In FFD, we set the operational threshold $\delta = \delta_{ideal} + \lambda_{margin}$, where $\lambda_{margin}$ acts as a safety buffer. A False Negative occurs if a token satisfies $s \geq m - \delta_{ideal}$ but is rejected by the scanner: $\hat{s} < \tilde{m} - \delta$.

Given $\hat{s} = s + e_{approx}$ and our definition of $\delta$, the rejection condition implies:

$$s + e_{approx} < \tilde{m} - (\delta_{ideal} + \lambda_{margin}) \tag{13}$$

Using $s \geq m - \delta_{ideal}$ and the fact that the Pseudo-Max $\tilde{m}$ is a lower bound on the true max $m$ ($\tilde{m} \leq m$), we obtain:

$$e_{approx} < (\tilde{m} - m) - \lambda_{margin} \leq -\lambda_{margin} \tag{14}$$

Thus, a False Negative requires the quantization error to exceed the safety margin in the negative direction. According to Theorem B.2, this probability is:

$$P(\text{False Negative}) \leq \exp \left( -\frac{\lambda_{margin}^2}{2\sigma^2 \alpha^2} \right) \tag{15}$$

This result provides a strong theoretical guarantee: the probability of missing a critical token decays exponentially with the square of the safety margin. In practice, setting $\delta = 5$ or $7$ provides a sufficiently large $\lambda_{margin}$ relative to the noise scale $\sigma$, explaining the near-perfect recall observed in our LongBench and RULER experiments despite the 2-bit representation.

## C. Bandwidth Proportionality of Quantization Depth

A central design choice in FFD is the 2-bit thumbnail with 8-bit residual ("2+8") configuration. To justify this choice empirically for the decode setting, we benchmark the decoding kernel latency under three quantization configurations—2+8, 4+8, and 8+8—using real text from the LongBench dataset on an RTX 4090. Table 4 reports the results.

The kernel speed is nearly proportional to the scan bandwidth: 2-bit scanning incurs one-quarter the I/O of 8-bit and half that of 4-bit. This proportionality is a distinctive characteristic of the decode setting where memory bandwidth dominates; it does not necessarily hold in prefill, where computation is the primary bottleneck and 4-bit may be a natural choice due to better hardware support for 4-bit arithmetic. This finding motivates the 2+8 design as the bandwidth-optimal configuration for long-context decoding.

*Table 4.* Decode kernel latency under different quantization depths (RTX 4090, 16k context).

| Configuration | Latency (ms/head) | Speedup vs. FlashAttn |
|---|---|---|
| 2+8 (FFD) | 0.105 | 10.65× |
| 4+8 | 0.201 | 5.60× |
| 8+8 | 0.406 | 2.82× |

## D. Head-Wise Variation under Fixed Delta

To assess whether FFD implicitly assumes uniform sparsity across attention heads, we conduct a controlled head-wise analysis on a sample from LongBench with $\delta = 5$ fixed. Across 1024 layer-head pairs (32 layers × 32 heads on Llama-3.1-8B), we measure the per-head keep ratio and salient hit rate. Table 5 summarizes the results.

*Table 5.* Head-wise variation statistics ($\delta = 5$, single LongBench sample, 1024 layer-head pairs).

| Statistic | Keep Ratio (%) | Salient Hit Rate (%) |
|---|---|---|
| Mean | 6.42 | 87.45 |
| Std. dev. | 14.14 | – |
| Min | 0.62 | – |
| Max | 100.0 | – |

The keep ratio spans from $0.62\%$ to $100\%$ with a standard deviation of $14.14$ percentage points, indicating that FFD does *not* impose a uniform sparsity structure. Rather, the same $\delta$-threshold rule allows different heads to exhibit markedly different retention behavior, with the mean salient hit rate remaining at $87.45\%$. Notably, both Quest and Twilight require retaining the first two layers' full-precision KV cache to avoid accuracy collapse, whereas FFD requires no such early-layer exception—suggesting that the content-aware scanning mechanism naturally captures the distributional properties that lead other methods to rely on architectural hand-tuning.

**Scope.** These statistics are derived from a single LongBench sample under a fixed $\delta = 5$. They should be interpreted as a mechanistic characterization rather than as claims about worst-case behavior across all task distributions.

## E. Failure-Case Profiling

To characterize when FFD's approximation is most stressed, we profile failure-prone conditions using two proxy metrics: the gap between the top-1 and top-2 attention scores (top1_top2_gap) and the standard deviation of attention scores (score_std). A layer-head pair is flagged as a *proxy-error* case if it falls into the lowest decile (bottom $10\%$) on either metric, and as a *high-error* case if it falls into the lowest decile on both. These are proxy indicators of flat or multi-peak attention distributions where quantization-based selection is most challenged.

We profile 3392 layer-head pairs across 4 attention dumps (2 models × 2 LongBench tasks). Table 6 summarizes the results.

*Table 6.* Failure-case profiling across 3392 layer-head pairs.

| Case Type | Proportion (%) | Mean Keep Ratio (%) | Mean Salient Hit Rate (%) |
|---|---|---|---|
| Regular | 80.84 | 2.43 | 63.74 |
| Proxy-error | 19.16 | 13.15 | 74.67 |

The dominant response to challenging distributions is *conservative retrieval*: the mean keep ratio rises from $2.43\%$ (regular) to $13.15\%$ (proxy-error), while the salient hit rate actually *improves* from $63.74\%$ to $74.67\%$. This confirms the structural asymmetry discussed in Section 5.2: when the pseudo-max approximation deviates from the true global maximum, the resulting threshold is more permissive (not more restrictive), so the selection defaults to retaining more blocks rather than

dropping salient ones. The primary consequence of these cases is therefore an efficiency penalty rather than an accuracy failure. We caution that proxy-error flags are based on attention-level statistics and have not been causally linked to downstream task errors.

## F. Cross-Architecture Generalization: Qwen2.5

Table 7 reports the full per-subtask RULER results on Qwen2.5-7B-Instruct.

*Table 7.* Qwen2.5-7B-Instruct RULER results (32k context, all 13 subtasks).

| Method | SK-1 | SK-2 | SK-3 | MK-1 | MK-2 | MK-3 | MQ | MV | VT | SQA | HOT | CWE | FWE | Avg |
|---|---|---|---|---|---|---|---|---|---|---|---|---|---|---|
| Base | 100 | 100 | 100 | 100 | 96 | 92 | 97.8 | 94.3 | 99.4 | 57 | 44 | 61.2 | 88.3 | 86.9 |
| KIVI | 100 | 100 | 86 | 99 | 54 | 7 | 88.5 | 86.3 | 94.4 | 49 | 40 | 49.3 | 82.7 | 72.0 |
| Quest | 100 | 100 | 97 | 98 | 68 | 4 | 96.3 | 92.8 | 99.6 | 39 | 37 | 36.1 | 67.3 | 71.9 |
| Stream. | 3 | 4 | 6 | 8 | 5 | 11 | 7.3 | 8.0 | 11.6 | 18 | 26 | 15.0 | 88.3 | 16.2 |
| Twilight | 100 | 96 | 91 | 85 | 86 | 44 | 66.3 | 73.3 | 80.4 | 44 | 35 | 54.3 | 82.7 | 72.1 |
| FFD ($\delta = 7$) | 99 | 99 | 99 | 99 | 95 | 87 | 98.3 | 93.5 | 98.6 | 53 | 40 | 68.4 | 87.0 | 85.9 |

Table 8 reports the LongBench breakdown. FFD maintains a clear advantage over all sparse baselines on Qwen, confirming generalization beyond Llama.

*Table 8.* Qwen2.5-7B-Instruct LongBench results (category averages).

| Method | S-Doc | M-Doc | Summ | Few-Shot | Syn | Code | Avg |
|---|---|---|---|---|---|---|---|
| Base | 24.35 | 27.88 | 24.35 | 59.50 | 32.99 | 11.71 | 30.13 |
| KIVI | 24.49 | 27.04 | 23.26 | 59.01 | 24.38 | 11.89 | 28.34 |
| Quest | 25.00 | 26.68 | 22.70 | 58.81 | 32.18 | 11.61 | 29.49 |
| Stream. | 15.92 | 22.51 | 18.96 | 52.53 | 5.48 | 13.29 | 21.45 |
| Twilight | 26.07 | 30.53 | 23.61 | 59.19 | 32.65 | 11.81 | 30.64 |
| FFD | 24.65 | 28.38 | 24.23 | 59.31 | 32.30 | 11.79 | 30.11 |

FFD maintains a clear accuracy advantage over sparse baselines on Qwen while generalizing beyond Llama without architecture-specific tuning. For Twilight, its stronger LongBench average is largely driven by M-Doc and Syn categories; however on RULER retrieval tasks, FFD retains a substantial lead across all key subtasks (e.g., MK-3: 87 vs. 44, MQ: 98.3 vs. 66.3).

## G. Component-wise Ablation Study

To disentangle the contributions of our three co-design axes—scan precision (FP16 vs. 2-bit), selection rule (top-$k$ vs. top-$\delta$), and execution path (split vs. fused+fullgraph)—we conduct a controlled ablation study under matched conditions. All variants are evaluated with a 16k prefill and 128 decode steps on an RTX 4090 (batch size 1). Table 9 reports the end-to-end throughput and downstream task accuracy for six configurations.

*Table 9.* Component-wise ablation of scan precision, selection rule, and execution path (16k context, RTX 4090).

| # | Variant (Scan – Rule – Execution) | Tok/s | RULER AVG | LongBench AVG |
|---|---|---|---|---|
| 1 | FP16 – top-$k$ – split | 17.43 | 86.82 | 25.87 |
| 2 | FP16 – top-$\delta$ – split | 21.64 | 90.39 | 25.90 |
| 3 | 2-bit – top-$k$ – split | 14.80 | 86.95 | 25.83 |
| 4 | 2-bit – top-$\delta$ – split | 24.65 | 90.40 | 25.90 |
| 5 | 2-bit – top-$k$ – fused | 21.21 | 86.95 | 25.83 |
| 6 | 2-bit – top-$\delta$ – fused (FFD) | **53.96** | **90.40** | **25.90** |

The ablation reveals three key findings. First, replacing top-$k$ with top-$\delta$ alone improves both accuracy (RULER: 86.82 $\rightarrow$ 90.39) and throughput (17.4 $\rightarrow$ 21.6 tok/s), confirming that top-$\delta$ achieves distribution-adaptive sparsity without the overhead of global synchronization (V1 vs. V2). Second, 2-bit scanning is not beneficial in isolation—under top-$k$, the quantized scan degrades throughput (V1 vs. V3: 17.4 $\rightarrow$ 14.8) due to compute-bound sorting overhead. However, when paired with top-$\delta$, which reduces selection to a scalar comparison, the reduced memory footprint of 2-bit becomes a strict asset (V2 vs. V4: 21.6 $\rightarrow$ 24.7). Third, and most importantly, the system-level fusion and CUDA Graph capture exhibit a **synergistic amplification** with the algorithmic components. While fusion alone yields a modest 1.4$\times$ gain under top-$k$ (V3 vs. V5), the same system optimizations deliver a 2.2$\times$ boost when combined with top-$\delta$ (V4 vs. V6), demonstrating that the algorithmic and system designs are inseparably linked.

## H. Component Accuracy Ablation on RULER

Table 10 provides the full per-subtask RULER breakdown for the four non-fused variants of the component ablation study (Section 4.5). The key observation is that CWE degrades severely under the fixed-budget (top-$k$) selector (scores of 24.6 and 25.8), whereas top-$\delta$ restores it to 68.2–69.5. This is because CWE requires the model to count word frequencies across the full context, a task that demands distribution-aware adaptive sparsity rather than a fixed token budget.

*Table 10.* Per-subtask RULER scores for component ablation variants (Llama-3.1-8B, 32K, split execution).

| Subtask | fp-fixed-split | fp-delta-split | q2-fixed-split | q2-delta-split |
|---|---|---|---|---|
| SK-1 | 100 | 100 | 100 | 100 |
| SK-2 | 100 | 100 | 100 | 100 |
| SK-3 | 100 | 100 | 100 | 100 |
| MK-1 | 97 | 98 | 97 | 97 |
| MK-2 | 97 | 99 | 98 | 99 |
| MK-3 | 99 | 99 | 96 | 99 |
| MQ | 98.5 | 99.0 | 97.8 | 99.0 |
| MV | 99.3 | 100 | 98.8 | 99.8 |
| VT | 98.6 | 98.2 | 98.4 | 98.0 |
| SQuAD | 67 | 68 | 69 | 68 |
| Hotpot | 54 | 54 | 56 | 55 |
| CWE | 24.6 | 68.2 | 25.8 | 69.5 |
| FWE | 93.7 | 91.7 | 93.7 | 91.0 |
| Avg | 86.8 | 90.4 | 87.0 | 90.4 |

