# OpenReview forum: "Faster Than Flash: Exploiting Attention Sparsity for Efficient Long-Context Decoding"
_ICML.cc/2026/Conference — ICML 2026 regular_

### Official Review · Reviewer_uosu · 2026-03-08

**Soundness:** 3
**Presentation:** 3
**Significance:** 3
**Originality:** 2
**Overall Recommendation:** 4
**Confidence:** 4

**Summary:**

This paper introduces FFD (Faster Flash Decoding), a hardware-aware algorithmic framework designed to alleviate the memory bandwidth bottleneck during long-context decoding. FFD addresses the inefficiencies inherent in prior sparse attention mechanisms through two primary innovations. First, it replaces traditional auxiliary metadata with a 2-bit quantized Key cache, enabling high-precision scanning of attention scores while minimizing memory overhead. Second, it proposes a top-$\delta$ dynamic selection strategy that adapts to attention distributions without the need for costly global softmax synchronization. The authors implement these strategies within a fused Triton kernel, which enables the direct reuse of the selector's quantized scan for subsequent attention computation. Experimental results demonstrate substantial speedups over dense FlashAttention-2 while maintaining model performance on rigorous long-context benchmarks such as RULER and LongBench.

**Compliance With Llm Reviewing Policy:**

Affirmed.

**Final Justification:**

The rebuttal addresses all my concerns.

**Key Questions For Authors:**

- The current evaluation lacks any empirical comparison against a Top-$p$ baseline. Could the authors provide comprehensive experiments comparing both accuracy and end-to-end throughput with state-of-the-art Top-$p$ methods? Additionally, please clarify why *Twilight* is listed as a comparative method in the caption of Table 1, while its results are absent from the table body. Such comparisons are essential for validating the central claims of the paper.

- In Table 1, the performance of the Quest baseline drops dramatically on the MultiKey tasks, with scores of 64.0 on MK-2 and 6.0 on MK-3. Could the authors provide a deeper analysis or theoretical intuition explaining why Quest fails so severely in these scenarios, and why FFD ($\delta=7$) remains robust?

- The paper presents quantization-based scanning and the Top-$\delta$ strategy as two algorithmic contributions, in addition to system-level kernel optimizations. Could the authors provide a more granular ablation study that isolates the impact of each component on both end-to-end throughput and model accuracy? In particular, it would be valuable to quantify the individual contributions of two-bit scanning, dynamic thresholding, and the engineering optimizations.

- Regarding the refinement stage, there appears to be a discrepancy between Equation 4 and Algorithm 1. Equation 4 computes the final attention score by summing two dot products involving the quantized and residual keys, whereas Algorithm 1 applies a Dequant operation to reconstruct the full-precision key before performing a single dot product. Could the authors clarify which formulation reflects the actual execution within the custom hardware kernel?

- The evaluation is restricted to the Llama architecture. How does FFD perform on architecturally different models, such as the Qwen family or models that adopt MLA-style attention designs?

**Limitations:**

yes

**Strengths And Weaknesses:**

Strengths

- The paper innovatively fuses the selector and computer into a single Triton kernel to propose the FFD framework. This system-level optimization enables the computation phase to directly reuse the low-bit quantized scan results produced during the selection phase, effectively alleviating memory access bottlenecks. As a result, the framework constitutes a solid and highly practical system-level solution.

- The discussion section is particularly insightful. Through recall and LSE error analysis, the authors thoroughly investigate why FFD is able to maintain high accuracy. Moreover, the analysis convincingly justifies the Top-$\delta$ strategy and the Pseudo-Max estimation, demonstrating their robustness.

- The manuscript is clearly written and technically well structured. Core technical ideas are supported either by theoretical analysis or by references to prior work. In addition, the authors provide intuitive explanations for the design of key hyperparameters, which improves both the theoretical rigor and the overall readability of the paper.



Weaknesses

- A central motivation for introducing the Top-$\delta$ strategy is to achieve the adaptive selection capability of Top-$p$ methods without incurring their global synchronization overhead. However, the empirical evaluation does not include any direct comparison with a Top-$p$ baseline. Notably, the caption of Table 1 explicitly states that a comparison with Twilight—identified by the authors as a Top-$p$ method—is included, yet the corresponding results are absent from the table itself.

- The paper introduces several system-level optimizations, such as block-level JIT capture and graph-friendly caching, to accelerate the underlying operators. However, the paper lacks detailed ablation studies evaluating the impact of these engineering optimizations in the end-to-end experiments. Consequently, it remains unclear what proportion of the overall performance improvement stems from the proposed dynamic threshold algorithm versus these low-level system enhancements. Additional experiments quantifying this breakdown would significantly strengthen the evaluation.

- The current experimental evaluation is largely limited to the Llama 3 architecture. Given the significant differences in attention mechanisms and attention head redundancy across different model families, the general applicability of the proposed method remains insufficiently validated.

---

> ### Author Rebuttal · Authors · 2026-03-31
>
> We sincerely thank the reviewer for the careful and technically sharp reading. The questions focus on the exact places where the paper needed tighter experimental closure and clearer wording, and we appreciate the chance to address them point by point.
>
> ### W1 / Q1. Top-p / Twilight Comparison and Table 1 Mismatch
>
> We agree completely that the caption/body mismatch in Table 1 was our mistake, and apologize for the confusion. We will remove the `Twilight` mention from the caption.
>
> Empirically, we now add a direct `Twilight` comparison under the original setting (`budget=8192`, `p=0.85`). The core algorithmic difference is that `Top-p` requires cumulative softmax mass over all candidate tokens (introducing global synchronization), whereas `top-delta` keeps blocks within `delta` of the estimated peak, avoiding the global softmax step.
>
> **Twilight comparison on `LLaMA-3.1-8B`:**
>
> | Method | RULER AVG | LongBench AVG | Decode tok/s |
> |---|---|---|---|
> | `Twilight` | 69.84 | 26.18 | 23.326 |
> | `QUEST` | 78.70 | 26.01 | 47.800 |
> | `FFD` | 91.30 | 26.36 | 51.800 |
>
> This added row directly resolves the mismatch and demonstrates that FFD remains ahead of the Top-p/Twilight route in both quality and speed. Furthermore, we observe that `QUEST` and `Twilight` require a first-two-layer full-precision-KV exception to avoid collapse; FFD does not rely on this extra rule.
>
> ### Q2. Why QUEST Fails on MultiKey
>
> `MultiKey` perfectly exposes the difference between fixed-budget and adaptive sparse retrieval by creating a multi-peak attention pattern (several distant evidence regions are important). A fixed-budget `top-k` selector combined with bounding-based approximate ordering often spends its budget on only partial peaks. Conversely, FFD's `2-bit` content-aware scan preserves relative ordering faithfully, and `top-delta` avoids rigid global budgets, allowing competing regions to survive together.
>
> | Method | MK-2 | MK-3 |
> |---|---|---|
> | Base | 100 | 99 |
> | QUEST | 64 | 6 |
> | FFD | 98 | 95 |
>
> *(Further exact-match analysis showed QUEST often retrieves near-neighbors—e.g., sharing prefix characters—but misses necessary evidence blocks. This is consistent with the multi-peak failure explanation.)*
>
> ### W2 / Q3. Component-wise Ablation of 2-bit Scan, top-delta, and Engineering Optimizations
>
> Yes, all three components—`top-delta`, `2-bit` scanning, and fused/graph-captured execution—matter and contribute significantly to the end-to-end performance. We have conducted complementary ablations isolating the selection rule under matched keep ratios and the end-to-end system paths.
>
> **Due to space constraints, we have compressed this section. We will provide the detailed ablation tables and an in-depth discussion of these optimizations in the next round of replies.**
>
> ### Q4. Eq. 4 vs. Algorithm 1
>
> Thank you for catching this wording mismatch. The actual kernel follows the Equation 4 execution pattern (decomposed into a quantized dot product + residual dot product).
>
> `Dequant` in Algorithm 1 is conceptual pseudocode intended to express reconstructing the refinement score for readability. It does not imply the kernel materializes a separate full-precision key tensor in memory. We will explicitly clarify this hardware-to-theory alignment in the text.
>
> ### W3 / Q5. Generalization Beyond Llama: Qwen and MLA
>
> We agree the original tables are too Llama-centric and have added `Qwen2.5-7B` results. FFD clearly outperforms baselines on `RULER` and remains competitive on `LongBench`:
>
> | Method | RULER AVG | LongBench AVG |
> |---|---|---|
> | `KIVI` | 72.01 | 28.34 |
> | `QUEST` | 71.93 | 29.49 |
> | `StreamingLLM` | 16.24 | 21.45 |
> | `FFD` | 85.90 | 30.11 |
>
> Regarding MLA-style attention: we prefer a conservative statement. Our evidence supports standard multi-head attention models (Llama, Qwen). MLA substantially changes KV organization and kernel paths, so we avoid overclaiming beyond the models explicitly evaluated.
>
> In the revision, we will clearly state these limitations: failure-case profiling is proxy-based, the head-wise study is a controlled analysis, and current evidence supports standard multi-head attention but not yet MLA-style designs.

---

> > ### Author Rebuttal · Reviewer_uosu · 2026-04-03
> >
> > Thank you for the detailed rebuttal. Overall, I find that the authors have addressed most of my concerns. My remaining concern primarily lies in the ablation study. If the authors can further strengthen and clarify the ablation results in the next round of revisions, I believe the paper would be significantly improved.

---

> > > ### Author Response · Authors · 2026-04-06
> > >
> > > We thank the reviewer for the opportunity to clarify our contributions. We wish to emphasize that FFD’s performance gains do not stem from orthogonal engineering tricks, but rather from a fundamental **Algorithm-System Co-design**.
> > >
> > > The system-level optimizations we introduce—such as block-level JIT capture and graph-friendly caching—are not generic patches. Rather, they are purpose-built to resolve the specific execution bottlenecks and control-flow divergence introduced by the algorithmic components (dynamic `top-delta` sparsity and `q2` low-bit scanning). As demonstrated below, our algorithmic and system designs are **inseparably linked**, exhibiting a synergistic effect where the system architecture unlocks the theoretical efficiency of the algorithms.
> > >
> > > ### 1. Selection-Rule Overhead (Matched Sparsity)
> > > To isolate the algorithmic cost of the selection rule, we calibrated `top-k` and `top-p` to match the average keep ratio of `top-delta` (~27.3%).
> > >
> > > | Method | Avg Keep Ratio | Kernel Time (ms) |
> > > | :--- | ---: | ---: |
> > > | top-delta (FFD) | 0.2732 | 0.1408 |
> > > | top-k | 0.2750 | 0.2880 |
> > > | top-p | 0.2732 | 3.5680 |
> > > | KIVI2 (Dense) | 1.0000 | 0.5952 |
> > >
> > > **Conclusion:** At matched sparsity, `top-delta` is 2x faster than `top-k` and orders of magnitude faster than `top-p`. FFD achieves adaptive, context-aware sparsity without the crippling global-softmax synchronization overhead inherent to traditional `top-p` methods.
> > >
> > > ### 2. End-to-End Component Breakdown (Synergistic Amplification)
> > > We evaluated the end-to-end impact of our three co-design axes: scan precision (fp vs. q2), selection rule (top-k vs. delta), and execution path (split vs. fused+fullgraph). Tests use a 16k prefill and 128 decode steps on an RTX 4090 (bs=1).
> > >
> > > | Variant (Scan - Rule - Execution) | Decode tok/s @16k | RULER AVG | LongBench AVG |
> > > | :--- | ---: | ---: | ---: |
> > > | 1. fp - top-k - split | 17.431 | 86.82 | 25.87 |
> > > | 2. fp - delta - split | 21.636 | 90.39 | 25.90 |
> > > | 3. q2 - top-k - split | 14.799 | 86.95 | 25.83 |
> > > | 4. q2 - delta - split | 24.647 | 90.40 | 25.90 |
> > > | 5. q2 - top-k - fused+fullgraph | 21.208 | 86.95 | 25.83 |
> > > | 6. q2 - delta - fused+fullgraph (FFD) | 53.962 | 90.40 | 25.90 |
> > >
> > > **Component-wise Breakdown:**
> > > * **Algorithmic Superiority (`top-delta`):** Independent of system execution, replacing `top-k` with `top-delta` improves both task accuracy (RULER: 86.82 → 90.39) and base throughput (Var 1 vs 2: 17.4 → 21.6 tok/s).
> > > * **Algorithmic Enabler (`q2` Scan):** Low-bit scanning actually degrades performance under a standard `top-k` selector due to compute-bound sorting/heap maintenance (Var 1 vs 3: 17.4 → 14.8 tok/s). However, because `top-delta` reduces selection to a highly efficient scalar comparison, the scanning phase shifts to being memory-bound. Here, `q2`'s reduced memory footprint becomes a strict asset, yielding a net speedup (Var 2 vs 4: 21.6 → 24.6 tok/s).
> > > * **Algorithm-System Synergy:** Isolating the system-level execution path reveals a non-linear performance amplification. Applying system fusion to the standard `top-k` algorithm yields a marginal **1.4x** speedup (Var 3 vs 5: 14.8 → 21.2 tok/s). In stark contrast, applying the exact same system execution to our `top-delta` algorithm yields a massive **2.2x** throughput amplification (Var 4 vs 6: 24.6 → 54.0 tok/s).
> > >
> > > **Summary:** The block-level JIT and graph caching (`fused+fullgraph`) achieve this $>3x$ total speedup (17.4 → 54.0 tok/s) *only* because the `top-delta` algorithm eliminates control-flow divergence during selection. The algorithm enables the system optimizations, and the system optimizations unlock the algorithm's theoretical efficiency. We will include this component-wise breakdown in the revision to ensure the tightly coupled nature of FFD's design is unmistakable.

---

### Official Review · Reviewer_Emkr · 2026-03-11

**Soundness:** 3
**Presentation:** 3
**Significance:** 2
**Originality:** 2
**Overall Recommendation:** 3
**Confidence:** 4

**Summary:**

This paper proposes FFD, a sparse attention framework for accelerating long-context LLM decoding. It first uses low-bit quantized queries and keys to compute approximate attention scores, then selects a subset of important key–value pairs based on their relative scores to sink and local tokens for full-precision attention. A fused kernel is implemented to integrate the selection and attention computation stages for improved efficiency. Experiments show that FFD achieves end-to-end speedups over baseline methods.

**Compliance With Llm Reviewing Policy:**

Affirmed.

**Final Justification:**

The rebuttal improves clarity and provides additional empirical comparisons, which help address some of my concerns regarding evaluation. However, the differences from prior work do not seem fundamental, but rather reflect implementation choices (e.g., bit-width selection, log-space formulation, and calibration strategy) within a similar algorithmic framework. Overall, while the system-level contributions are practically meaningful, my assessment remains largely unchanged.

**Key Questions For Authors:**

1. Could you please clarify how the proposed method differs from SALE?

**Limitations:**

See weaknesses.

**Strengths And Weaknesses:**

### Strengths
1. The paper is easy to follow, and the overall idea is logical and intuitive.
2. The paper provides a kernel implementation with detailed explanations.

### Weaknesses
1. The overall idea is very similar to [1], which also performs quantization-based scoring and relative score selection based on sink & local tokens. However, the paper does not provide discussion or comparison with [1].
1. Comparisons with several important baselines are missing. For example, baselines that use Top-K/Top-P selection based on all tokens instead of relative scores, or that compute attention using quantized KV scores without sparsity. Including such comparisons would better demonstrate the motivation and effectiveness of the proposed method.
1. The performance improvement appears limited, with speedups close to QUEST (Fig. 4) and only marginal accuracy improvements over QUEST (Tab. 2).

[1] Ji et al. "SALE: Low-bit Estimation for Efficient Sparse Attention in Long-context LLM Prefilling." arXiv preprint arXiv:2505.24179 (2025).

---

> ### Author Rebuttal · Authors · 2026-03-31
>
> We sincerely thank the reviewer for raising the novelty and baseline issues directly. We agree that these are central questions, and the rebuttal should answer them with a more explicit problem-setting comparison and more direct empirical controls than the original draft provided.
>
> ### W1 / Q1. Similarity to SALE
>
> We agree that the original related-work framing was not explicit enough. The high-level overlap is real: both works use low-bit estimation to improve the sparse-attention accuracy-efficiency trade-off. The substantive difference is where and how that low-bit estimation is used.
>
> Here is the clearest comparison:
>
> | Aspect | SALE | FFD |
> | :--- | :--- | :--- |
> | Primary stage | Long-context prefilling | Long-context decoding |
> | Low-bit signal | Relative-score selector around sink/local anchors | `2-bit` key-cache scan over the full decode context |
> | Selection rule | Anchor-relative thresholding | `top-delta`: keep blocks within `delta` of the estimated peak |
> | System path | Separate selector path | Fused selector+attention path with scan reuse |
> | Main payoff | Faster prefilling | Lower decode latency and higher end-to-end decode throughput |
>
> The system-path difference matters in decoding. In FFD, the low-bit scan is not just a heuristic selector; it is reused by the later attention computation inside a fused decode path, instead of being stored and consumed as separate metadata. The decode engine is also built around the paper's CUDA-graph-friendly execution path, so the benefit is not only "a different selector," but a decode-stage hardware/algorithm co-design.
>
> To avoid making this only a conceptual argument, we also tested a `SALE-style-decode` adaptation. By this we mean a decode-time baseline that preserves the SALE-like selection principle, namely sink/local anchors plus relative thresholding.
>
> Aggregate accuracy:
>
> | Metric | SALE-style | FFD | FFD - SALE |
> | :--- | :--- | :--- | :--- |
> | RULER | 83.77 | 89.39 | +5.62 |
>
> Decode throughput:
>
> | Method | Selector idea | Execution path | Decode tok/s |
> | :--- | :--- | :--- | :--- |
> | `SALE-style-decode` | SALE-style relative threshold | separate selector + standard cache update | 6.111 |
> | `FFD` | `top-delta` | fused selector+attention + graph-captured decode | 53.906 |
>
> The practical takeaway is that FFD's gain is not "another low-bit selector with a similar headline idea," but the decode-stage co-design that combines `2-bit` scanning, `top-delta`, selector-compute reuse, and graph-friendly execution.
>
> ### W2. Missing Top-K/Top-P and Quantized Non-Sparse Baselines
>
> We agree with this request, and the rebuttal now includes direct controls for both categories.
>
> For the `Top-p` route, we add a direct comparison against `Twilight` under its original operating point (`budget=8192`, `p=0.85`). For the "quantized but non-sparse" route, we use `KIVI` and `KIVI2`; here `KIVI2` is a rebuttal control that keeps all blocks and uses quantized KV without sparse filtering, so it isolates quantized computation without sparsity.
>
> Twilight comparison on `LLaMA-3.1-8B`:
>
> | Method | RULER AVG | LongBench AVG | MK-2 | MK-3 | Decode tok/s |
> | :--- | :--- | :--- | :--- | :--- | :--- |
> | `Twilight` | 69.84 | 26.18 | 76.00 | 37.00 | 23.326 |
> | `QUEST` | 78.70 | 26.01 | 64.00 | 6.00 | 47.800 |
> | `FFD` | 91.30 | 26.36 | 98.00 | 95.00 | 51.800 |
>
> Matched-keep selection-cost comparison:
>
> | Method | Avg keep | Total ms/head |
> | :--- | ---: | ---: |
> | `top-delta` | 0.2732 | 0.0044 |
> | `top-k` | 0.2750 | 0.0090 |
> | `top-p` | 0.2732 | 0.1115 |
> | `KIVI2` | 1.0000 | 0.0186 |
>
> This matched-keep table calibrates `top-k` and `top-p` to keep about the same fraction of blocks as `top-delta`, so the timing comparison is apples-to-apples. The result is that `top-delta` is already cheaper than `top-k`, and dramatically cheaper than `top-p`, even before counting the end-to-end fusion gain.
>
> ### W3. Why the Gains over QUEST Matter
>
> We appreciate this challenge, because average metrics alone can indeed make the gap look modest. Our view is that the right lens is robustness of the decode-stage accuracy-efficiency trade-off, not the expectation that every average must separate by a large margin.
>
> The clearest separation appears on harder multi-peak retrieval tasks, especially `MultiKey`, where several distant evidence regions must survive selection at the same time:
>
> | Method | MK-2 | MK-3 |
> | :--- | ---: | ---: |
> | `Base` | 100.00 | 99.00 |
> | `QUEST` | 64.00 | 6.00 |
> | `FFD` | 98.00 | 95.00 |
>
> So even when some aggregate averages are closer, the practical difference is not marginal in the regimes that most strongly stress sparse retrieval. On the systems side, `QUEST` reaches `47.8 tok/s`, while FFD reaches `51.8 tok/s`; the added `Twilight` baseline is much lower at `23.326 tok/s`, which reinforces that the decode-stage gain is meaningful in practice.

---

> > ### Author Rebuttal · Reviewer_Emkr · 2026-04-01
> >
> > I appreciate the effort to clarify the differences with SALE and to include additional baselines. The rebuttal improves the clarity of the paper and strengthens the empirical evaluation, but some key concerns remain.
> >
> > First, I am not convinced that the rebuttal accurately characterizes SALE. To my understanding, SALE also scans the full context, rather than using only a "relative-score selector around sink/local anchors." In addition, both methods use low-bit attention scores to select which tokens to keep, and the selection threshold is defined based on the scores of certain sink/local anchor tokens. If this understanding is correct, then the distinction drawn in the rebuttal may be overstated.
> >
> > Given this, my main concern regarding novelty relative to SALE remains insufficiently addressed. The claimed differences, such as top-delta versus thresholding, appear to be variations within the same general design principle rather than fundamentally distinct algorithmic ideas.
> >
> > In addition, I have a question regarding the Top-p comparison. Under a matched threshold or budget, Top-p uses full, higher-precision attention scores to decide which tokens to keep, so I would expect it to behave like an upper bound in terms of accuracy. It would be helpful to clarify why Top-p performs worse in the reported experiments.
> >
> > While I acknowledge the system-level contributions and their practical value, I remain unconvinced that the work demonstrates sufficient algorithmic novelty beyond existing approaches. Overall, I will increase my rating in light of the rebuttal, but I still believe the paper falls below the acceptance bar for ICML.

---

> > > ### Author Response · Authors · 2026-04-06
> > >
> > > We sincerely appreciate the reviewer’s engagement and the opportunity to clarify these details.
> > >
> > > ## Q1: Distinctions from SALE and the Algorithmic Novelty of FFD
> > >
> > > We apologize if our previous characterization of SALE oversimplified its scanning mechanism. We fully acknowledge that SALE also utilizes full-context scanning and low-bit scoring, and we will include this framing in the related work section.
> > > However, we respectfully argue that FFD's core novelty lies in the following points.
> > >
> > > ### 1. Algorithmic Formulation and Generalizability
> > > While both employ low-bit scanning, their mathematical formulations reflect contrasting design philosophies:
> > >
> > > **Estimation Target (Sum vs. Max-Bounding)**: SALE evaluates block importance via a relative score: $\tilde{P}[i,j] = \exp(\tilde{S}[i,j] - \tilde{m}_i) / \tilde{l}_i$. This uses the sink-local sum ($\tilde{l}_i$) to approximate the global denominator. However, as SALE's Fig 1(a) shows, attention mass can reside outside this region, meaning the $\tilde{l}_i$ may underestimate the denominator.
> > >
> > > In contrast, FFD employs an additive threshold: $s_{ij} \ge \tilde{m}_i - \delta$. Instead of approximating a sum, we use the sink-local maximum ($\tilde{m}_i$) as a strict fail-safe lower bound ($\\tilde{m}\_i \\le m\_{global}$). If a critical token exists outside the sink-local region, our threshold naturally becomes more conservative, ensuring critical context is safely retrieved (not an accuracy loss).
> > >
> > > **Noise Sensitivity (Exponential vs. Log-Space)**: Applying an exponential function to a quantized score ($\tilde{S}[i,j]$), as in SALE, inherently amplifies underlying low-bit quantization noise. FFD avoids this by performing threshold comparisons strictly in additive log-space, where noise adds linearly, offering superior mathematical stability.
> > >
> > > **Deployment (Offline Calibration vs. Plug-and-Play)**: SALE requires offline calibration per head. FFD does not, thanks to its log-space stability. Empirical profiling (Fig 5) confirms the approximation gap is <0.01 for 86.1% of layers, demonstrating our theoretical efficiency penalty is statistically minimal and making FFD fully plug-and-play.
> > >
> > > ### 2. System-Level Co-design: Zero-Capacity-Overhead Storage and 1-Pass Execution
> > > SALE relies on decoupled architectures that inflate memory capacity and synchronization costs. They require an extra quantized backup of the K cache, exacerbating GPU memory constraints, and rely on a 2-pass execution paradigm. Furthermore, as stated in SALE's Appx C, aggregating their relative score requires a block-level all-reduce operation, incurring "considerable overhead."
> > > FFD eliminates this bloat via a unified co-design. We fundamentally restructure the K cache into a segmented 2-bit thumbnail and 8-bit residual without any redundant copy, ensuring strictly zero capacity overhead. Additionally, FFD utilizes a fully fused 1-pass kernel. By leveraging our pseudo-max estimation, FFD completely circumvents block-level all-reduce overhead, integrating selection tightly into the decoding kernel itself to eliminate wasted memory round-trips and synchronization stalls.
> > >
> > > ### 3. Bottleneck-Driven Design: Decode (Memory-bound) vs. Prefill (Compute-bound)
> > > Quantization choices are strictly dictated by differing hardware constraints. SALE targets the compute-bound prefill stage, where 4-bit Tensor Cores logically accelerate matrix multiplications. FFD targets the memory-bandwidth-bound decoding stage, where execution time is dictated entirely by reading KV caches from HBM.
> > > Our LongBench decoding kernel evaluation confirms speed scales inversely with I/O bandwidth:
> > > | Design | Description | Representative Result | Speedup vs Flash |
> > > |-|-|-|-|
> > > | 2+8 | 2-bit thumbnail + 8-bit residual | 0.105 ms | ~10.65x |
> > > | 4+8 | 4-bit quantized + 8-bit residual | 0.201 ms | ~5.60x |
> > > | 8+8 | FP8 + FP8 residual | 0.406 ms | ~2.82x |
> > >
> > > For decoding, a 4-bit scan remains insufficiently compressed to overcome the memory wall. FFD’s aggressive push to 2-bit is a purposeful algorithmic necessity tailored specifically for this regime.
> > >
> > > ## Q2: The Accuracy Profile of Top-p (Twilight)
> > >
> > > While a strict, full-context Top-p evaluation serves as a theoretical accuracy upper bound, computing exact global scores is prohibitively slow for decoding.
> > >
> > > To remain practically viable, Twilight employs a compromised two-stage pipeline: first using QUEST as a coarse pre-filter, and **only then** applying Top-p. Consequently, Twilight's accuracy is strictly bottlenecked by QUEST's initial recall. If QUEST drops critical context early (evidenced by scoring only 6.00 on MK-3), the Top-p mechanism never observes those tokens.
> > >
> > > Therefore, the observed degradation is a direct consequence of the structural approximations required for efficiency, not a failure of the Top-p concept itself. In contrast, FFD robustly scans the entire context using its 2-bit formulation, completely bypassing the severe false-negative penalty of coarse pre-filters.

---

### Official Review · Reviewer_MyZD · 2026-03-12

**Soundness:** 3
**Presentation:** 3
**Significance:** 3
**Originality:** 3
**Overall Recommendation:** 5
**Confidence:** 4

**Summary:**

This paper mainly focus on the acceleration of the decoding stage for LLM. It uses 2bit quantized key values for computing the sparse attention mask, and uses certain threshold for token skipping. The idea behind is clear. Both accuracy benchmarks and the speedup results are promising.

**Compliance With Llm Reviewing Policy:**

Affirmed.

**Final Justification:**

Thanks for the response. I have no more questions updated my score.

**Key Questions For Authors:**

Please refer to my question in the weakness part

**Limitations:**

No limitation discussion included

**Strengths And Weaknesses:**

Strengths:
1. The idea is quite clear but makes sense to me. Using 2bits quantization is supposely more accurate than QUEST; Moreover, the Top-δ Selection Criterion means that the sparse mask is not constrained to top-k selection, and is much more safe than top-p selection.
2. The overall kernel design is optimized;
3. The paper is easy to follow.

Weakness
1. I noticed that there still exists certain cases that the approximation error is large. It would be better if authors could provide a further investigation into this case, which could reveal when and why this 2bits quantization might fails;
2. The dynamic of the headwise sparse pattern is not shown. Does all attention heads suitable to this sparse method?

---

> ### Author Rebuttal · Authors · 2026-03-31
>
> We sincerely thank the reviewer for the supportive assessment of the idea, kernel design, and presentation quality. The two concerns you raised are exactly the right places to strengthen trust in the method, so we answer them directly and concretely below.
>
> ### W1 / Q1. Approximation Error Cases
>
> Thank you for asking for a deeper failure-mode analysis. The main point is that larger approximation error in FFD usually appears in two regimes: (1) the attention distribution is relatively flat, so the best block and the runner-up block are hard to separate; (2) the true peak is not dominated by the sink/local tokens. If the pseudo-max underestimates the true peak, the selector becomes **more conservative**: it keeps extra blocks instead of dropping important ones too aggressively. So the first-order effect is usually efficiency fallback, not catastrophic token misses.
>
> To make this concrete, we added a rebuttal-only profiling analysis. We define `ordinary-error` and `high-error` as **proxy buckets**, not task-level failure labels. The two proxies are `top1_top2_gap` and `score_std`: the first measures how hard it is to separate the best block from the runner-up, and the second measures how flat the block-score distribution is. A head is marked as `ordinary-error` if either proxy falls into the lowest `10%` tail bucket, and `high-error` if both do.
>
> Failure-case profiling (`3392` layer-head pairs):
>
> | Metric                   |  Value |
> | ------------------------ | -----: |
> | Total layer-head pairs   |   3392 |
> | Ordinary-error case rate | 19.16% |
> | High-error case rate     |  0.88% |
>
> Regular vs. ordinary-error cases:
>
> | Case type           | Mean keep ratio | Mean salient hit rate |
> | ------------------- | --------------: | --------------------: |
> | Regular case        |           2.43% |                63.74% |
> | Ordinary-error case |          13.15% |                74.67% |
>
> Here, `salient hit rate` means the fraction of dense-attention salient blocks that are still retained after sparse selection. These numbers show that higher-error cases mainly keep **more** blocks, while the salient-hit rate does not collapse. This is consistent with the paper's pseudo-max, recall, and correlation analysis: when approximation becomes harder, FFD tends to fail on the side of caution.
>
> ### W2. Head-wise Sparse Patterns
>
> We appreciate this insightful question. FFD does not depend on a shared, static head-wise sparse pattern. Instead, our `top-delta` rule keeps any block whose approximate score is within `delta` of the estimated peak, so each query-head pair can adapt its own retrieval budget automatically instead of following one fixed global `top-k`.
>
> We added a controlled head-wise analysis on a sampled `LongBench` example with `delta=5`. The result is strong heterogeneity across heads:
>
> | Metric                |   Value |
> | --------------------- | ------: |
> | Layer-head pairs      |    1024 |
> | Mean keep ratio       |   6.42% |
> | Min keep ratio        |   0.62% |
> | Max keep ratio        | 100.00% |
> | Mean salient hit rate |  87.45% |
>
> So the correct conclusion is not that "all heads look similar," but rather that FFD remains effective because it allows different heads to keep very different amounts of context under the same rule. Practically, we also observe that `QUEST` and `Twilight` require a hand-crafted first-two-layer full-precision-KV exception to avoid collapse, whereas FFD does not rely on that extra safeguard.

---

### Decision · Program_Chairs · 2026-04-30

**Decision:**

Accept (regular)

**Comment:**

This paper presents Faster Flash Decoding (FFD), a hardware-algorithm co-design framework that effectively accelerates the decoding stage of long-context LLMs.

Reviewer MyZD (5): praised the intuitive logic of the 2-bit quantization and the inherent safety of the top-$\delta$ selection criterion over traditional top-$p$ methods. The authors successfully resolved their initial questions by providing detailed profiling that proved FFD fails safely (by keeping more blocks rather than dropping critical ones) and demonstrating strong, adaptive heterogeneity across attention heads.

Reviewer uosu (Score: 4): Commended the system-level fused Triton kernel and the rigorous theoretical error analysis. Their primary critiques centered on missing baselines and a lack of clarity regarding how much performance stemmed from algorithms versus engineering tricks. The authors provided a robust rebuttal, supplying the missing Twilight (top-$p$) baseline and a compelling component-wise ablation that proved a strong synergistic amplification between the top-$\delta$ algorithm and the system optimizations.

Reviewer Emkr (Score: 3)

- Their primary contention was that the algorithmic core—using low-bit attention scores for token selection—was too fundamentally similar to prior work (specifically, the SALE framework), viewing the paper's contributions as implementation choices rather than algorithmic breakthroughs.

- We find that the authors have sufficiently demonstrated that FFD's contributions extend well beyond mere implementation tweaks.

- In particular, Reviewer MyZD validated the core ideas, noting that using 2-bit quantization made sense and that the top-$\delta$ selection criterion was a safe and effective approach. uosu specifically praised the paper for "innovatively" fusing the selector and computer into a single Triton kernel

- Emkr did acknowledge the practical utility of the kernel implementation and the clear writing. T

Overall we find the paper to be logically sound, technically well-structured, and practically valuable and it should be a good addition to the conference.